# Measurement of Physical Fitness and 24/7 Physical Activity, Standing, Sedentary Behavior, and Time in Bed in Working-Age Finns: Study Protocol for FINFIT 2021

**DOI:** 10.3390/mps5010007

**Published:** 2022-01-13

**Authors:** Pauliina Husu, Henri Vähä-Ypyä, Kari Tokola, Harri Sievänen, Ari Mänttäri, Sami Kokko, Kaisu M. Kaikkonen, Kai Savonen, Tommi Vasankari

**Affiliations:** 1The UKK Institute for Health Promotion Research, 33500 Tampere, Finland; henri.vaha-ypya@ukkinstituutti.fi (H.V.-Y.); kari.tokola@ukkinstituutti.fi (K.T.); harri.sievanen@ukkinstituutti.fi (H.S.); ari.manttari@ukkinstituutti.fi (A.M.); tommi.vasankari@ukkinstituutti.fi (T.V.); 2Faculty of Sport and Health Sciences, University of Jyväskylä, 40014 Jyväskylä, Finland; sami.p.kokko@jyu.fi; 3Department of Sports and Exercise Medicine, Oulu Deaconess Institute Foundation, 90100 Oulu, Finland; kaisu.kaikkonen@odl.fi; 4Kuopio Research Institute of Exercise Medicine, 70100 Kuopio, Finland; kai.savonen@uef.fi; 5Department of Clinical Physiology and Nuclear Medicine, Science Service Center, Kuopio University Hospital, 70210 Kuopio, Finland; 6The Faculty of Medicine and Health Technology, Tampere University, 33014 Tampere, Finland

**Keywords:** accelerometer, health, noncommunicable diseases, register, adults, active travel, fitness

## Abstract

Background: Population studies gathering measured data on fitness and physical behavior, covering physical activity, standing, sedentary behavior, and time in bed, are scarce. This article describes the protocol of the FINFIT 2021 study that measures fitness and physical behavior in a population-based sample of adults and analyzes their associations and dose–response relationships with several health indicators. Methods: The study comprises a stratified random sample of 20–69-year-old men and women (*n* = 16,500) from seven city-centered regions in Finland. Physical behavior is measured 24/7 by tri-axial accelerometry and analyzed with validated MAD-APE algorithms. Health and fitness examinations include fasting blood samples, measurements of blood pressure, anthropometry, and health-related fitness. Domains of health, functioning, well-being, and socio-demographics are assessed by a questionnaire. The data are being collected between September 2021 and February 2022. Discussion: The study provides population data on physical fitness and physical behavior 24/7. Physical behavior patterns by intensity and duration on an hour-by-hour basis will be provided. In the future, the baseline data will be assessed against prospective register-based data on incident diseases, healthcare utilization, sickness absence, premature retirement, and death. A similar study will be conducted every fourth year with a new random population sample.

## 1. Introduction

Global disease burden has continued to shift away from communicable to noncommunicable diseases and from premature death to years lived with disability [1]. Strong evidence shows that physical inactivity, i.e., not meeting the current physical activity (PA) recommendation for health [2], increases the risk of many adverse health conditions, including coronary heart disease, type 2 diabetes, several cancers, anxiety and depression, and cognitive impairments, and shortens life expectancy [3]. According to a recent report, there is a curvilinear dose–response between the time spent in moderate-to-vigorous PA (MVPA) and the risk of all-cause mortality, with a lower risk associated with a higher time spent in MVPA [4]. Despite these established benefits of PA, a significant proportion of adults do not adhere to MVPA recommendations [5,6,7]. When measured by an accelerometer, the compliance is substantially lower than according to self-reports [6]. Both the economic costs and disease burden (disability-adjusted life years) of physical inactivity are substantial [8].

The current PA recommendation for health [2] also includes sedentary behavior (SB), a distinct behavior from physical inactivity defined as any waking behavior characterized by an energy expenditure less than 1.5 metabolic equivalents (MET) while in a sitting or reclining posture [9]. Accumulating evidence suggests that SB is a risk factor for all-cause and cardiovascular mortality, incident cardiovascular disease, and type 2 diabetes [10]. Population studies based on accelerometer data suggest that SB and MVPA are independently associated with all-cause mortality among less-active adults, but not among those engaged in higher levels of MVPA [11]. MVPA confers the largest health benefits [3] and risk reduction for cardiovascular disease mortality [11].

Light-intensity physical activity (LPA) is also included in the current PA recommendation for health [2]. It covers over one-fifth of the waking hours [12] and contributes substantially to the overall daily energy expenditure [13]. Data from population studies indicate that LPA is beneficially associated with obesity and markers of lipid and glucose metabolism [14]. LPA seems to be inversely associated with the all-cause mortality risk independent of MVPA [15,16]. Especially, frequent short bouts of LPA may have acute beneficial physiological effects, particularly among individuals with metabolic impairments [17].

It has been stated that PA of any intensity reduces the risk of premature death, while different intensities can be combined in numerous ways [18]. Replacing SB with PA of any intensity will produce health benefits, but the greatest benefits will occur when SB is replaced with MVPA [11]. Moreover, when SB is replaced with MVPA instead of LPA, the comparable health benefits will be achieved within less time [4].

Besides inactivity, low cardiorespiratory fitness (CRF) is also a strong predictor of all-cause mortality [19], cardiovascular diseases and events [19,20], type 2 diabetes [21], and various cancers [22,23]. An increment of one MET (resting metabolic rate 3.5 mL of oxygen per kilogram of body weight per minute) in CRF has been associated with a considerable (12–16%) reduction in all-cause, cardiovascular disease, and cancer mortality [24].

Studies measuring CRF, PA, and SB have shown that a high amount of SB is marginally associated with a lower CRF independent of PA [25]. Thus, a combination of decreasing SB and increasing PA, resulting in improved CRF, seems most beneficial for health [25,26]. However, the number of population-based studies including both the assessment of physical fitness (CRF) and accelerometer-measured PA and SB are yet rare [12,27,28].

In addition to being physically inactive or having prolonged bouts of SB or both, short and/or poor-quality sleep is a prominent feature of modern cultures [29]. Physical inactivity and a high amount of SB may contribute to poor sleep, while the combined health effect of PA, SB, and sleep may be stronger than either alone [29]. Sleep duration has a U-shaped association with body mass index, diabetes, cardiovascular diseases, and metabolic syndrome [30] Poor sleep is also associated with several health problems, including inflammation and impaired cognition [31]. Regular exercise seems to confer beneficial effects on sleep time and quality [32]. Especially, moderate PA seems to be associated with better sleep quality [33]. Both lifestyle PA and MVPA seem to have a large positive effect on the relationship between sleep and cardio-metabolic health, including waist circumference, blood pressure, and fasting insulin concentration [34].

Within a 24 h time, physical behavior forms a continuum from time in bed (TIB) intending to sleep to vigorous PA [35]. Given the finite 24 h period in a circadian cycle, TIB, standing, SB, and PA at different intensities are co-dependent, meaning that a change in one category leads to a change in at least one of the other categories [35,36]. Thus, compositional analysis of the physical behavior over the entire 24 h period provides a solid foundation for a statistically valid and comprehensive investigation of the associations between the relative distribution of time spent in different physical behaviors [35,36,37]. However, current ways to monitor daily physical behavior with device-based methods are quite variable in terms of the measurement or wear-time, device placement, and analysis algorithms used. This variance has been recognized as one factor that feeds current controversies in the epidemiology of daily physical behavior [38]. Chastin et al. [4] reported that a hip-worn accelerometer is more likely to reflect the relationships for waking day behaviors than wrist-worn devices. Therefore, it has been recently proposed that using a hip-worn accelerometer during waking hours and a wrist-worn accelerometer when intending to sleep might provide an optimal setup for measuring physical behavior over a 24 h time [38].

The present article describes the study protocol of the population-based FINFIT 2021 study. The main purpose of this cross-sectional study with a random population-based sample is to assess physical fitness and 24/7 physical behavior among working-age adults in Finland. Both independent and combined associations of fitness and intensity-specific PA, standing, SB, and TIB, and their dose–response relationships to several measured, self-reported, or register-based indicators of health will be analyzed. The first FINFIT study was conducted in 2017–2019 [12] to develop and verify the protocol for the FINFIT 2021 study. In the future, a new random sample of 20–69-year-old adults is planned to be recruited every fourth year.

The FINFIT 2021 study is part of a larger consortium, Healthy Lifestyles to Boost Sustainable Growth (STYLE) (www.styletutkimus.fi/en/frontpage/ accessed on 11 November 2021). The STYLE project will exploit the results of the FINFIT 2017 [12] and 2021 population studies as a baseline status of adults. The aim of the STYLE project is to promote active traveling (walking or cycling for transport) in the cities of Helsinki, Tampere, Turku, and Jyväskylä.

## 2. Materials and Methods

The Finnish Ministry of Education and Culture creates public policy and programs for continuous national surveillance of measured, population-level physical fitness, PA, SB, and sleep among all age-groups. This provides the rationale for conducting the FINFIT 2021 study among working-age adults.

The FINFIT 2021 is a cross-sectional population study based on a stratified random sample of 20–69-year-old Finnish men and women from seven city-centered regions (Helsinki, Turku, Tampere, Jyväskylä, Kuopio, Oulu, and Rovaniemi). Potential participants are drawn from a population census: from seven regions, 150 men and women in five age groups (20–29, 30–39, 40–49, 50–59, and 60–69) are drawn, adding up to a total of 10,500 individuals. Additionally, 1500 individuals are drawn from the Helsinki region and 500 from 9 municipalities from the Tampere and Turku regions. This will sum up to a total of 16,500 individuals. Invitation letters containing information about the study and informed consent, with an option to withdraw from the study at any time, are being mailed to potential participants.

The study measurements include (1) health and fitness examinations at the local research centers; (2) accelerometer-measured PA, standing, SB, and TIB 24/7 over one week (7 consecutive days); and (3) a health-related questionnaire either online or in paper form. The participants are free to choose whether they are willing to take part in all three parts of the study or only the latter two. The participants can indicate their choice via an online system, and via that, they are also able to settle an appointment at the local research center. When attending the appointment, before any examinations, the participants return a signed informed consent to research assistants. If no appointment is chosen, the participants return the signed informed consent via mail. Thus, most of the participants have only one appointment during which they return the informed consent, give blood samples, participate in health-screening, perform the fitness tests, receive the accelerometer for 24/7 measurement, and receive the study questionnaire. If the fasting blood sample cannot be taken during the first appointment, very few participants will have a second appointment for that. Those participants who are not willing to attend the appointment at all can receive the accelerometer and questionnaire by mail. After the accelerometer measurement week is over, the participant returns the accelerometer to the research center by mail using a pre-paid envelope.

The recruitment and examinations started in September 2021 and will continue until February 2022. The study regions and the contents are presented in Figure 1. The Regional Ethics Committee of the Expert Responsibility Area of Tampere University Hospital gave ethical approval for the study (R21050).

The conceptual design of the study is based on the modified version of the Toronto model on Physical Activity, Fitness and Health [39]. The main measurements of the FINFIT 2021 study are listed according to this model in Figure 2.

The health-related contents of the FINFIT 2021 questionnaire are described in Figure 2: perceived health [40], diagnosed diseases, use of health services and sickness absences, musculoskeletal symptoms, mobility difficulties [41,42], work ability [43,44], work-related recovery, day-time tiredness, sleepiness and amount of night sleep, psychological well-being [45], and quality of life [46]. In addition, data from several national registers including attendance and cost of health care, medications, sickness absences, premature retirement, and death will be used both in cross-sectional analyses and in later prospective follow-up studies. These registries include Care Register for Health Care, Register of Primary Heath Care visits, National Causes of Death Register, Statistics on reimbursements for medical expenses, and Statistics on sickness allowances.

## 3. Procedures

### 3.1. Assessment of 24 h Physical Behavior (PA, Standing, SB, TIB)

A tri-axial accelerometer (RM42, UKK Terveyspalvelut Oy, Tampere, Finland) is worn on the right hip during waking hours and on the nondominant wrist during TIB. All participants are asked to wear the accelerometer for seven consecutive days, except during water-based activities. Those who participate in the health and fitness examination receive both oral and written instructions for the use of the accelerometer, and they start to wear the accelerometer just before the fitness testing. Those not attending the examination receive the accelerometer and written instructions by mail.

The accelerometer collects and stores the raw triaxial data in actual g-units within a ±16 g range at a 100 Hz sampling rate. The resolution of the 13-bit accelerometer is 3.9 mg. The accelerometer is initialized so that it starts collecting data if the absolute value of the difference between a reference value and the incident acceleration exceeds 187.5 mg in any axis, and if, during the next five seconds, the difference exceeds 500 mg in any axis; if not, the accelerometer returns to a quiescent state [47]. Whenever the raw acceleration in any axis exceeds the previous limits, the reference values are updated with the incident acceleration values. If a continuous quiescent time is longer than 120 min [48], this period is considered a nonwear time. Otherwise, the quiescent time denotes the stationary time. In the case of detected nonwear time, the given measurement day is excluded from the data.

Regarding PA analysis, the mean amplitude deviation (MAD) is determined from the resultant acceleration of three orthogonal acceleration components in six-second epochs [49]. The MAD is a valid (R^2^ = 0.94) indicator of incident oxygen consumption during locomotion over a wide range of speeds [50]. The epoch-wise MAD values are converted to METs (3.5 mL/kg/min of oxygen consumption) and further smoothed by calculating an exponential moving average of each epoch time point [51]. The epochs are classified into three intensity categories both in absolute and relative terms [51]. The absolute thresholds are 1.5–2.9 MET for LPA, 3.0–5.9 MET for moderate PA (MPA), and at least 6.0 MET for vigorous PA (VPA) [52]. The relative thresholds, based on the individual oxygen uptake level reserve (VO_2_R), are less than 40% of VO_2_R for LPA, 40–60% of VO_2_R for MPA, and at least 60% of VO_2_R for VPA [52]. In addition, the daily step count number is measured [53].

SB and standing are identified for the epochs, where the predicted MET value is less than 1.5. The accelerometer orientation in terms of the vector of Earth’s gravity is taken as the reference, and the angle for posture estimation (APE) is determined from the incident accelerometer orientation in relation to the reference vector [53]. Posture is classified as standing if the APE value is less than 11.6°, sitting if the value is between 11.6° and 30.0°, reclining if the value is between 30.0° and 73.0°, and lying if the value is greater than 73.0°. In free-living conditions, about a 90% agreement in identifying SB was observed between the results from the hip- and thigh-worn accelerometers [53]. The number of breaks in SB, called sit-to-stand transitions, are calculated as the number of SB (lying, reclining, and sitting) periods during which the one-minute exponential moving average of the estimated MET value indicates no movement and which ends up with a clear vertical acceleration followed by a standing position or movement [53,54].

The location of the accelerometer is automatically recognized from the profile of detected changes in its orientation. For the hip location, the changes are inherently smaller than in the wrist. If the accelerometer is turned upside down, it is more likely worn on the wrist. Similarly, if there are only small changes in the orientation, the accelerometer is more likely worn on the hip [12].

The TIB algorithm is based on an open-access algorithm [55], which monitors changes in the wrist orientation between consecutive epochs and calculates the time interval between changes that exceed five degrees. The TIB is classified into three categories according to the number of changes in the wrist angle during the time window covering the preceding 10 min and following 10 min periods [12]: high-movement (HM), medium-movement (MM), and low-movement (LM) periods.

A summary of parameters of PA, standing, SB, and TIB is described in Table 1. In addition, accumulated times representing varying bout lengths of SB, standing, and PA at different intensity levels will be determined. In addition, individual weekly and daily peak MET levels will be calculated using different lengths of the exponential moving average filters [51].

### 3.2. Self-Reported Sleep

The participants also keep a sleep diary during the seven nights they are using the accelerometer on the wrist. The following data are gathered after waking up in the morning: (1) time having gone to bed (hour; min); (2) number of times getting out of bed during sleeping; (3) wake-up time (hour; min); as well as (4) alertness assessed with the Karolinska Sleepiness Scale in 9-point Likert scale [56,57]: 1 = very alert; 3 = alert; 5 = neither alert nor sleepy; 7 = sleepy (but not fighting sleep); 9 = very sleepy (fighting sleep).

### 3.3. Self-Reported Physical Activity and Sedentary Behavior

A questionnaire on participants’ PA in terms of frequency, duration, and type of activity is employed. There are also questions about walking, cycling, and other physically active commuting as well as sitting hours during a working and nonworking day. The major PA- and health-related contents of the questionnaire are presented in Figure 2.

### 3.4. Assessment of Health-Related Physical Fitness

Physical fitness is assessed by the following reliable, feasible, and valid field-based tests of health-related fitness depending on the age of the participant: shoulder-neck mobility, jump and reach, modified push-up [58,59,60], and six-minute walk test [61,62] for all participants, and one-leg stand for the participants aged at least 45 years. Before the fitness tests, a standard pre-testing health screening [59] is conducted. The screening includes measurements of weight and height, waist circumference, blood pressure, and self-reported PA level. Educated fitness testers employ screening results to exclude participants with minor health limitations from certain fitness tests. In the case of severe disease or symptoms, the participant is excluded from all tests [59].

### 3.5. Assessment of Health

The health assessment includes fasting blood samples taken after a 12 h overnight fasting. The later biochemical analyses will include serum levels of total cholesterol, high- and low-density lipoprotein cholesterol, triglyceride, and indicators of glucose metabolism (fasting plasma glucose and HbA1c). After taking the blood samples and measurement of blood pressure, the participants are offered a light snack. The health examination ends with pre-testing health screening [59], which is described above as part of the fitness testing protocol.

### 3.6. Statistical Analysis

Clustering (city-centered regions), stratification (sex and age groups), and weighting will be defined in the design of the complex survey procedure and considered when estimating population characteristics.

To obtain valid population estimates, each participant will be given a selection probability weight based on a few key population variables (e.g., age and sex). Weighting adjusts for differences between the survey data and the actual population. Along with the traditional statistical analysis, the compositional analysis will be used to account for the co-dependency between the parameters of physical behavior (PA, standing, SB, and TIB) and indicators of health.

## 4. Expected Results

The FINFIT 2021 study is one of the first population-based studies of working-age adults that measures physical behavior 24/7 by an accelerometer together with several components of physical fitness, as well as self-reported and register-based data on health indicators. The main purpose of the study was to investigate the independent and combined associations of physical behavior and fitness with health outcomes. The present approach allows the evaluation of differences among these factors in demographically different population groups, and for the first time, it will elaborate on their possible synergistic associations with health.

It has been hypothesized that a combination of decreased SB and increased PA, resulting in improved CRF, is the most beneficial for health [63]. Longevity and health may be promoted by undertaking physical activities across the full intensity spectrum, including unstructured LPA [18]. The magnitude of risk reduction is three times greater when PA is assessed by accelerometry instead of self-reported methods [18]. In short, the associations between physical behavior and health may be mediated through fitness while being modified by heredity and other factors related to lifestyle, personality, and bio-psychosocial environment [39]. The 24/7 physical behavior offers an interesting opportunity to assess behavioral compositions in relation to the health indicators of interest.

The strengths of the study include the exploration of multivariable models concerning the complex nature of the highly prevalent noncommunicable diseases in a large population sample of adults. Determining the associations between physical behavior, fitness, and health will provide new evidence and guidance for health professionals and the community to devise individually more tailored behavioral interventions to reduce the risk of diseases and promote public health. In addition, the previous data from the FINFIT 2017 study gathered with identical methods from a similar adult population from the same geographic areas [12] provide a unique opportunity to verify the associations that will be found in the FINFIT 2021 study.

The main risk of the study pertains to possibly low participation rates of the randomly selected sample, especially concerning the selected subgroups related to geographic area, socioeconomic status, PA, or health. In the FINFIT 2017 study, the main weakness was the selective participation, where only 42% of the sample was reached, and 51% of the reached persons agreed to participate in the study. [12] Of the 10,500 persons invited, 2256 were considered as eligible participants with valid accelerometer data (at least 4 days, 24 h per day). Most (78%) of these eligible participants used the accelerometer for 6–7 days per week. However, the participants were more likely to be married and more educated than the general population in Finland [12]. Based on these experiences, the participation rate may remain relatively low and also indicate selection in the FINFIT 2021 study. Further, the participation rate may be affected by the continuing COVID pandemic. To overcome this risk, the health and fitness examination is voluntary, and the accelerometer and questionnaire can be delivered to the participant without face-to-face contacts by mail if needed. We consider it important that population-based studies on physical behavior and related health associations are also executed during the exceptional pandemic time.

Other risk factors include potential health-related incidents or risks occurring during the fitness testing, loss of accelerometers during 24/7 measurements, and incompletely filled questionnaires. To manage these risks, based on the previous experiences from the FINFIT 2017 study, we send reminder messages to those participants who have not responded to the initial invitation. Before the health examination, we conduct an individual pre-testing health-screening for each participant to identify potential health risks related to fitness tests. Participants can also interrupt the testing anytime. To facilitate the return of the accelerometers, the participants receive a prepaid envelope with written instructions, as well as a reminder message and phone call if needed. To enhance the filling rate of the questionnaires, the participants can choose between paper and online forms. In addition, in the case of a few weeks of no-response, they receive a reminder message.

The FINFIT 2021 study provides numerous health-related variables and accelerometer-measured PA, standing, SB, and TIB together with different components of fitness. In the future, the baseline data will be assessed against prospective register-based data on incident diseases, healthcare utilization, sickness absence, and premature retirement and death. A similar study will be conducted every fourth year with a new random population sample.

## Figures and Tables

**Figure 1 mps-05-00007-f001:**
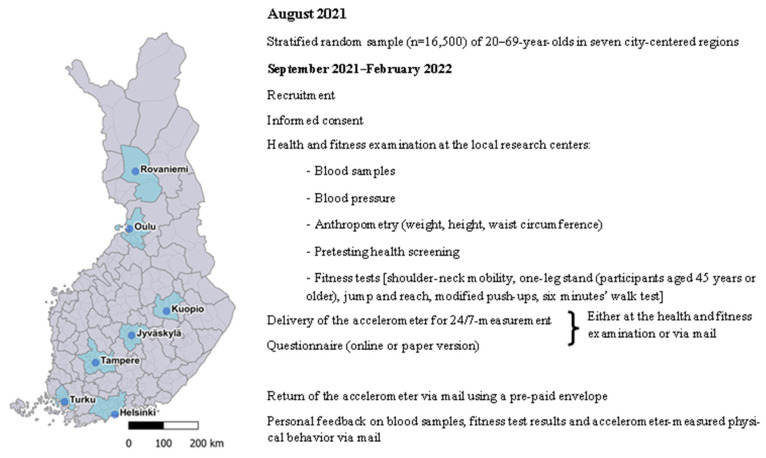
The seven study regions and flowchart of the data collection. Municipality and city region area boundaries, Statistics Finland, Tilastointialueet 2017. The material was downloaded from the Statistics Finland’s interface service on 11 November 2021 with the license CC BY 4.0.

**Figure 2 mps-05-00007-f002:**
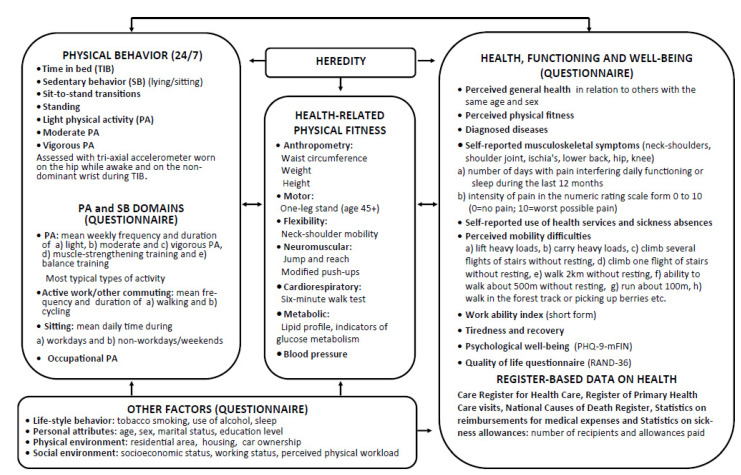
Assessment methods used in the FINFIT 2021 study.

**Table 1 mps-05-00007-t001:** Accelerometer-derived parameters of the physical behavior.

Time in Bed = Wrist-Worn Time	Hip-Worn Time
Number (N) of over 5° Changes in the Wrist Angle during 20 min Window	Physical Behavior	Absolute Threshold	Relative Threshold	APE (°)
High movement	N ≥ 14	Lying	MET ≤ 1.5	MET ≤ 1.5	73≥
Medium movement	2 ≤ N ≤ 13	Reclining	MET ≤ 1.5	MET ≤ 1.5	30–73
Low movement	N ≤ 1	Sitting	MET ≤ 1.5	MET ≤ 1.5	11.6–30
		Standing	MET ≤ 1.5	MET ≤ 1.5	0–11.6
		Light PA	1.5 < MET ≤ 3	1.5 < MET & VO_2_R ≤ 40%	
		Moderate PA	3 < MET ≤ 6	40% < VO_2_R ≤ 60%	
		Vigorous PA	MET > 6	VO_2_R > 60%

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
