# Peer review of "Measurement of Physical Fitness and 24/7 Physical Activity, Standing, Sedentary Behavior, and Time in Bed in Working-Age Finns: Study Protocol for FINFIT 2021"

_mps, 2022, doi:10.3390/mps5010007_

Round 1

Reviewer 1 Report

The protocol is well structured and the aim of the study is well defined. 

The project is ambitious because centered on a large number of subjects. These kinds of studies, for their complexity, often had an elevated number of drop-outs. However, it will be useful to evaluate the adherence to the protocol and the compliance of the subjects.

In my opinion, sleep and physical activity levels should be monitored for at least one week and not only for 24 h. 

Author Response

Dear reviewer,

Thank you very much for your valuable comments. The comments were carefully checked through and the following list will clarify the modifications made. In the revised manuscript file, the changes are marked with MS word track changes. We hope that these revisions improve the quality of our manuscript. 

Comment 1: The protocol is well structured and the aim of the study is well defined. 

  • Response 1: Thank you!

Comment 2: The project is ambitious because centered on a large number of subjects. These kinds of studies, for their complexity, often had an elevated number of drop-outs. However, it will be useful to evaluate the adherence to the protocol and the compliance of the subjects.

  • Response 2: Thank you for this comment. We quite agree that an elevated number of drop-outs may be the main challenge in this study. It is obvious that this kind of study may have a low response rate, especially during the Covid-19 pandemic. However, it is important that population-based studies are executed also during the pandemic time since the pandemic is supposed to influence physical activity, sedentary behavior and sleep of the participants. Covid-19 pandemic has been added as one of the risk factors of the study on lines 365-370.
  • We will carefully evaluate adherence and compliance of the participants. Response rate and number of the participants will be reported in each age and sex subgroup as a part of the results. However, we will not receive data about the reasons for non-participation. We have also addressed this topic by describing our experience in the previous FINFIT 2017 study on lines 357-364.
  • To obtain valid population estimates, each of the participants will be given a selection probability weight based on few key population variables (e.g., age and sex). Weighting adjusts for differences between the survey data and the actual population as indicated on lines 317-319.

Comment 3: In my opinion, sleep and physical activity levels should be monitored for at least one week and not only for 24 h. 

  • Response 3: We are sorry for causing this misunderstanding. Certainly sleep, physical activity and sedentary behavior will be monitored for one week, 24 hours for seven consecutive days. This has been mentioned in several parts of the manuscript and a clarification has been added on lines 146-147 and 160. In our previous FINFIT 2017 study 2256 participants (39.4%) wore the accelerometer at least 4 days for 24 hours per day, which was the acceptance criterion for valid data collection. These eligible participants represented 95% of people who were basically willing to use the accelerometer. Most (78%) of these eligible participants used the accelerometer for 6–7 days, and 97% of them had at least one weekend day in the data [12, Husu et al. 2021].

In addition to these revisions, the order of the references has been updated and small language corrections made to the figures.

Sincerely,

Pauliina Husu (corresponding author)

Reviewer 2 Report

Dear authors,

thank you very much for the presentation of the plannend/starting study. The number of participants is very ambitious and is distributed over seven centers, what is the minimum number to be achieved for the study?

How large was the group size of the FINFIT 2017 study?

Line 182 Does every participant participate here or is it a subgroup? This is not entirely clear in the wording.

How many appointments do participants have to go through all the studies?

The authors plan to recruit new participants every four years, is it also planned to follow up participants longitudinally?

Please check the english wording e.g. line 82 better write rare instead of scare.

Author Response

Dear reviewer,

Thank you very much for your valuable comments. The comments were carefully checked through and the following list will clarify the modifications made. In the revised manuscript file, the changes are marked with MS word track changes. We hope that these revisions improve the quality of our manuscript. 

Dear authors,

thank you very much for the presentation of the plannend/starting study.

Comment 1: The number of participants is very ambitious and is distributed over seven centers, what is the minimum number to be achieved for the study?

  • Response 1: Thank you. Based on our experiences from the FINFIT 2017 study, the participation rate is likely to remain relatively low also in the FINFIT 2021 study. We expect that at least 2.000 participants will participate. This has been discussed on lines 357-365. However, we understand that the participation rate may be influenced by the Covid-19 pandemic, but simultaneously we consider that it is very important to investigate physical activity, sedentary behavior, and sleep among working-aged adults during this long-lasting and exceptional pandemic time. This has been discussed on lines 365-370.

Comment 2: How large was the group size of the FINFIT 2017 study?

  • Response 2: The sample size of the FINFIT 2017 study was 10 500 leading to 2256 eligible participants for the accelerometer measurements. This has been specified on lines 359-360.

Comment 3: Line 182 Does every participant participate here or is it a subgroup? This is not entirely clear in the wording.

  • Response 3: All participants are invited to take part in the accelerometer measurement. They can receive the accelerometer and related instructions either at the health and fitness examination or remotely by mail. The wording has been clarified on lines 200-205.

Comment 4: How many appointments do participants have to go through all the studies?

  • Response 4: Most of the participants have only one appointment during which they return the informed consent, give blood samples, participate in health-screening, perform the fitness tests, receive the accelerometer for 24/7 measurement and receive the study questionnaire. If the fasting blood sample cannot be taken during the first appointment, very few participants will have a second appointment for that. Those participants who are not willing to attend the appointment at all can receive the accelerometer and questionnaire by mail. After the accelerometer measurement period (one week) is over, the participant returns the accelerometer to the research center by mail using a pre-paid envelope. This has been clarified on lines 153-161.

Comment 5: The authors plan to recruit new participants every four years, is it also planned to follow up participants longitudinally?

  • Response 5: Thank you for asking clarification for this. Yes, we will take a random sample of Finnish working-age adults every fourth year. We are planning to follow-up the participants via population registries as indicated on lines 188-194. No follow-up measurements are planned for this study.

Comment 6: Please check the english wording e.g. line 82 better write rare instead of scare.

  • Response 6: Thank you for pointing this out. This has been corrected (line 86) as suggested. In addition, several other language corrections have been made throughout the text.

In addition to these revisions, the order of the references has been updated and small language corrections made to the figures.

Sincerely,

Pauliina Husu (corresponding author)